# Cardiovascular Signal Entropy Predicts All-Cause Mortality: Evidence from The Irish Longitudinal Study on Ageing (TILDA)

**DOI:** 10.3390/e24050676

**Published:** 2022-05-11

**Authors:** Silvin P. Knight, Mark Ward, Louise Newman, James Davis, Eoin Duggan, Rose Anne Kenny, Roman Romero-Ortuno

**Affiliations:** 1The Irish Longitudinal Study on Ageing (TILDA), School of Medicine, Trinity College Dublin, D02 R590 Dublin, Ireland; wardm2@tcd.ie (M.W.); louise.newman@tcd.ie (L.N.); davisj5@tcd.ie (J.D.); dugganeo@tcd.ie (E.D.); rkenny@tcd.ie (R.A.K.); romeroor@tcd.ie (R.R.-O.); 2Discipline of Medical Gerontology, School of Medicine, Trinity College Dublin, D02 R590 Dublin, Ireland; 3Mercer’s Institute for Successful Ageing (MISA), St. James’s Hospital, D08 E191 Dublin, Ireland; 4Global Brain Health Institute, Trinity College Dublin, D02 PN40 Dublin, Ireland

**Keywords:** sample entropy, mortality, cardiovascular, blood pressure, TILDA

## Abstract

In this study, the relationship between cardiovascular signal entropy and the risk of seven-year all-cause mortality was explored in a large sample of community-dwelling older adults from The Irish Longitudinal Study on Ageing (TILDA). The hypothesis under investigation was that physiological dysregulation might be quantifiable by the level of sample entropy (SampEn) in continuously noninvasively measured resting-state systolic (sBP) and diastolic (dBP) blood pressure (BP) data, and that this SampEn measure might be independently predictive of mortality. Participants’ date of death up to 2017 was identified from official death registration data and linked to their TILDA baseline survey and health assessment data (2010). BP was continuously monitored during supine rest at baseline, and SampEn values were calculated for one-minute and five-minute sections of this data. In total, 4543 participants were included (mean (SD) age: 61.9 (8.4) years; 54.1% female), of whom 214 died. Cox proportional hazards regression models were used to estimate the hazard ratios (HRs) with 95% confidence intervals (CIs) for the associations between BP SampEn and all-cause mortality. Results revealed that higher SampEn in BP signals was significantly predictive of mortality risk, with an increase of one standard deviation in sBP SampEn and dBP SampEn corresponding to HRs of 1.19 and 1.17, respectively, in models comprehensively controlled for potential confounders. The quantification of SampEn in short length BP signals could provide a novel and clinically useful predictor of mortality risk in older adults.

## 1. Introduction

In 2015 it was estimated that 8.5% of the world’s population were aged 65 years and older, a proportion projected to increase to 13% by 2030, and further to 16.7% by 2050; this would be equivalent to an average annual increase of 27.1 million older people from 2015 to 2050 [1]. As we age, dysregulation of cardiovascular and neurovascular physiological functions is thought to be associated with adverse health outcomes and premature death. Dysregulation of these systems may be quantified in short-length, peripherally measured data using signal entropy, with recent work from our group showing this measure to be associated with pre-disability frailty status, longitudinal cognitive performance, and accelerated brain ageing [2,3,4].

The concept of neurocardiovascular instability (NCVI) refers to abnormal neural control of the cardiovascular system, which may adversely affect the dynamic behavior of blood pressure (BP) [5]. Older individuals are more prone to NCVI due to age-related physiological changes in the cardiovascular system, cerebral blood flow, autonomic nervous system (ANS), and humoral (hormonal) system. The ANS is responsible for controlling the body’s visceral functions and maintaining homeostasis [6], with one of the key ANS mechanisms, the baroreceptor reflex, being responsible for regulating short-term control of systemic BP [5]. NCVI could increase the risk of mortality through associated alterations in end-organ structure and function potentially deriving from impaired BP control. Understanding the mechanisms of NCVI and its potential causal association with mortality risk is critical, as early detection and treatment of related conditions could prevent premature mortality. However, measures of NCVI based on BP that capture this risk are lacking. We hypothesized that abnormalities in these physiological control mechanisms may be detectable and quantifiable by the level of disorder in short-length continuously measured BP signals.

Disorder in physiological signals can be assessed by means of signal entropy [7]. Entropy is a measure of irregularity/unpredictability, with lower entropy values assigned to periodic, predictable data, and higher entropy values to irregular, unpredictable data. Multiple implementations of entropy have been proposed for the analysis of time-varying physiological signals including approximate entropy (ApEn), sample entropy (SampEn), multi-scale entropy, and cross-entropy [8,9,10,11,12]. In this study we used SampEn, given its advantages over ApEn, namely the fact that self-matches are not counted and that resulting measures are largely independent of sample length [11]. Briefly, SampEn is defined as the negative natural logarithm of the conditional probability that two trajectories of length *m* remain similar for *m* + 1, within a tolerance specified as ±r× standard deviation (SD) of the timeseries of length *N*. However, for health research applications, different ways of processing continuously measured beat-to-beat (BtB) BP signals prior to SampEn calculation may yield different results. Previous works have investigated SampEn calculations from ‘beat-domain’ BP [2,3,4,13], ‘time-domain’ BP (incorporating both BtB BP and cardiac interval series (CIS) data via interpolation) [2,3,4], and CIS approaches [13,14,15]; in the present work we aim to investigate the utility of all three approaches with regard to mortality prediction. In addition, it is important to know how SampEn of BP signals compare with the more traditional heart rate variability (HRV) and resting heart rate (RHR) measures, which have been previously associated with mortality risk in older adults [16,17,18,19].

Thus, in this study we investigate associations between SampEn calculated in continuous non-invasively peripherally measured BP data and all-cause mortality over a seven-year period in a large sample of community-dwelling older adults. In doing so, we compare different processing approaches for BtB BP data prior to the calculation of SampEn (time-domain at 5 Hz, beat-domain, and CIS). We investigate these approaches across a range of *m* and *r* values, and also compare the ability of these different entropy measures to predict mortality with HRV and RHR measures.

## 2. Materials and Methods

### 2.1. Study Population

This research was carried out as part of an ongoing nationally representative prospective cohort study of community-dwelling older adults; The Irish Longitudinal Study on Ageing (TILDA) [20,21]. TILDA collects information on the health, economic and social circumstances of people aged 50 years and over. Wave 1 of the study (baseline) took place between October 2009 and February 2011 (n = 8507, with most assessments conducted in 2010), and subsequent data was collected approximately every two years over four longitudinal waves (wave 2: February 2012 to March 2013; wave 3: March 2014 to December 2015; wave 4: January to December 2016; wave 5: January to December 2018). Waves 1 and 3 included a comprehensive health assessment conducted at a dedicated health assessment center; waves 2, 4, and 5 were non-health center assessment waves. The full cohort profile has been previously described in detail [20,21]. Ethical approval was granted for each wave from the Faculty of Health Sciences Research Ethics Committee at Trinity College Dublin, Dublin, Ireland, and all participants provided written informed consent. All research was performed in accordance with the Declaration of Helsinki.

### 2.2. Cardiovascular Measurements

At wave 1 health assessment, blood pressure waveforms were measured continuously at 200 Hz using a Finometer MIDI device (Finapres Medical Systems BV, Amsterdam, The Netherlands) and recorded via a 12-bit resolution analogue-to-digital converter. All measurements were carried out at an ambient temperature (21 to 23 °C) in a comfortably lit room. Participants lay supine, and after a stabilization period, five minutes of data were collected. Data from the last minute of supine rest (i.e., resting state: RS) was utilized for the main analyses presented in this study, to naturally maximise data stationarity as much as possible. However, results are also presented for the full five minutes of RS data, using 5 Hz ‘time-domain’ data (produced by interpolating the BtB BP data and CIS data together into a 5 Hz time-series), beat-domain data (i.e., BP values evenly spaced at each successive beat), and CIS. Signals for BtB systolic blood pressure (sBP) and diastolic blood pressure (dBP) were extracted using MATLAB (R2020b, The MathWorks, Inc., Natick, MA, USA). Of note, as in our previous work [2,3], the temporal positioning of all beats was maintained to within the 5 Hz temporal resolution using the time-domain method, i.e., the data were not simply artificially extended by inserting a prescribed number of ‘artificial points’ between beats, a method which has been previously (and correctly) recommended against [13]. Data were transformed using the method proposed by Tarvainen et al. to detrend and increase stationarity; this method is based on a smoothness priors approach, and operates like a time-varying finite-impulse response high-pass filter [22]. A surface 3-lead electrocardiogram (ECG) was also continuously recorded at 4 kHz using a Medilog AR12 system (Schiller, Baar, Switzerland) during the same five-minute period of supine rest, to allow for the calculation of RHR and HRV measures.

### 2.3. Entropy Analysis

Entropy analysis was performed on the sBP and dBP data in MATLAB, using freely available code [23]. A detailed description of the algorithms used to compute SampEn has been previously reported in detail [11]; however, below we provide a brief overview.

For a time series of length *N*, $B_{i}^{m}\left(r\right)$ is defined as the number of template vectors of length *m*, $x_{m}\left(j\right)$, similar to $x_{m}\left(i\right)$ (within *r*) divided by *N* − *m* − 1, where *j* = 1...*N* − *m*, with *j* ≠ *i* (to avoid self-matches). The average $B_{i}^{m}\left(r\right)$ for all *i* is given as
(1)$$B^{m}\left(r\right)=\frac{1}{N-m}\overset{N-m}{\underset{i=1}{\sum }}B_{i}^{m}\left(r\right).$$

Similarly, we define $A_{i}^{m}\left(r\right)$ as the number of template vectors of length *m* + 1, $x_{m+1}\left(j\right)$, similar to $x_{m+1}\left(i\right)$ (within *r*) divided by *N* − *m* − 1, where *j* = 1...*N* − *m*, with *j* ≠ *i*. The average $A_{i}^{m}\left(r\right)$ for all *i* is given as
(2)$$A^{m}\left(r\right)=\frac{1}{N-m}\overset{N-m}{\underset{i=1}{\sum }}A_{i}^{m}\left(r\right).$$

SampEn was then calculated as
(3)$$\mathrm{SampEn}\left(m,r,N\right)=-\ln\left(\frac{A^{m}\left(r\right)}{B^{m}\left(r\right)}\right).\;$$

In this study, a range of *m* (embedding dimension; the length of the data segment being compared) values were investigated (*m* = [1–5]). Similarly, a range of *r* (similarity criterion) values were also investigated (*r* = [0.1, 0.15, 0.2, 0.25, 0.3, 0.35, 0.4, 0.45, 0.5, 0.55, 0.6]). SampEn was calculated for all potential combinations of *m* and *r*, for both sBP and dBP, with one-minute of 5 Hz time-domain data, five-minutes of 5 Hz time-domain data, five-minutes of beat-domain data, and CIS data.

### 2.4. RHR and HRV Analyses

ECG data were firstly band-pass filtered between 0.01 and 1000 Hz, and a proprietary algorithm was then used to detect the R peak of each heart beat recorded on the signal [24]. Linear interpolation was utilized to exclude supra-ventricular ectopic beats and noise from the data. All recordings were screened for atrial fibrillation (AF) using criteria from the European society of cardiology [25], and those identified with AF were subsequently excluded from the RHR/HRV analyses. Other arrhythmias were also detected and excluded by the Darwin software system. Five-minute epochs of RR interval data were analyzed. The mean RHR (beats-per-minute (bpm)) was calculated for this period, as well as the standard deviation of NN intervals (SDNN, ms). Frequency domain (FD) features were calculated from spectral estimates derived using an autoregressive (Burg method) parametric algorithm, with the assignment of 256 discrete frequency bins. FD features were derived by integrating the power spectral density across two frequency bands, namely: low frequency power (LF, 0.04–0.15 Hz, ms^2^) and high frequency power (HF, 0.15–0.4 Hz, ms^2^). SDNN, LF, and HF measures were corrected for the average RR interval prior to analyses [26]. It is thought that HF measures are reflective of parasympathetic activity while LF measures reflect both sympathetic and parasympathetic activity [19].

### 2.5. Mortality Data Linkage

The participants’ date of death was identified from official death registration data and linked to their TILDA survey and health assessment data. Linking was performed for all individuals who died between April 2010 and March 2017. Full details of the data linkage procedures are described elsewhere [27].

### 2.6. Covariates

As part of the TILDA survey, the following self-reported measures were also recorded and included as covariates in the fully adjusted models reported herein: age, sex, educational attainment, number of cardiovascular conditions (angina, high blood pressure, heart failure, heart murmur, abnormal heart rhythm, heart attack, high cholesterol), diabetes, alcohol consumption habits (CAGE) [28], smoking history, and antihypertensive medication use (coded using the Anatomical Therapeutic Chemical Classification (ATC): antihypertensive medications (ATC C02), diuretics (ATC C03), β-blockers (ATC C07), calcium channel blockers (ATC C08), and renin-angiotensin system agents (ATC C09)). Additionally, anthropometric measurements of height to the nearest 0.01 m (Seca 240 Stadiometer, Seca Ltd., Birmingham, UK) and weight to the nearest 0.1 kg (Seca 861 Electronic Scales, Seca Ltd., Birmingham, UK) were measured during the health center assessment, and body mass index (BMI) was calculated from the formula weight [kg]/(height [m])^2^. In order to account for the likely non-linear relationship between BMI and mortality [29], BMI was stratified by ‘underweight/normal’ (BMI < 25), ‘overweight’ (25 ≤ BMI < 30), ‘obese’ as (30 ≤ BMI < 35), and ‘morbidly obese’ (BMI ≥ 35), as per World Health Organization (WHO) guidelines [30].

### 2.7. Statistical Analysis

Statistical analysis was performed using STATA 15.1 (StataCorp, College Station, TX, USA). Descriptive statistics were performed with calculation of the mean and SD of continuous variables and count and percentage of categorical variables and differences in means and frequencies were evaluated by means of the t-test and chi-squared test, respectively. Cox proportional hazards regression models were utilized to estimate the hazard ratios (HRs) with 95% confidence intervals (CIs) for the association between BP SampEn and all-cause mortality. Respondents lost to follow-up were right-censored at the end of the follow-up-period (31 March 2017). The bivariate relationships between SampEn (time-domain, beat-domain, and CIS) and mortality risk, using a range of *r* and *m* values, were calculated and explored graphically, and these results informed the *r* and *m* choice for the main models used subsequently. Three sets of models were used: (i) unadjusted (bivariate); (ii) adjusted for age, sex, and education; and (iii) fully adjusted for the following covariates: age, sex, education, BMI, antihypertensive medication, diabetes, number of cardiovascular conditions, smoking status, and alcohol consumption. To compare the relative performance of the different entropy approaches outlined above (time-domain, beat-domain, and CIS), as well as the more traditional cardiovascular measures (RHR, SDNN, LF and HF), with regards the prediction of all-cause mortality risk, all measures were standardized (z-scores). Non-normally distributed measures of HRV (SDNN, LF and HF) were log transformed prior to standardization and analysis.

## 3. Results

### 3.1. Participant Characteristics

In total, 8507 participants were recruited at wave 1 of TILDA, 5035 of whom were aged over 50 years and attended a health center assessment at wave 1 (baseline). Adequate cardiovascular data were available for 4543 individuals for the calculation of baseline (wave 1) SampEn values (mean (SD) age: 61.9 (8.4) years; 54.1% female). Compared to the non-deceased cohort, on average, participants in the deceased cohort were 8.5 years older and proportionally this cohort contained 14.1% less females. The deceased cohort were less educated, had a higher proportion of antihypertensive medication use, had almost twice the prevalence of diabetes, had a greater number of cardiovascular conditions, and had a higher proportion of past or current smokers. Average SampEn measures were significantly higher in the deceased cohort, compared with the non-deceased, for both time-domain and CIS approaches, however, not for the beat-domain approach to data processing. Full exclusions are illustrated in Figure 1 and full baseline demographics for all three cohorts are provided in Table 1.

### 3.2. Effects of ‘m’ and ‘r’ Paramater Choice on Mortality Prediction and Mean SampEn Values

Figure 2 shows the results from Cox proportional hazards regression models investigating bivariate associations between SampEn calculated from time-domain (5 Hz, 60 s and 300 s; a–d), beat-domain (e,f), and CIS (g,h) sBP and dBP data and seven-year all-cause mortality, across a range of *m* and *r* values. Overall, *m* = 1 provided the highest HRs and most-consistently low *p*-values, particularly for the time-domain data. For all time-domain data, HRs were consistently significant when using *m* = 1 and *r* ≥ 0.15 (see Figure 2a–d). Overall, the beat-domain approach provided the least consistent results with regard to mortality prediction, with significance only being reached with the dBP data for very specific *m* and *r* combinations. Furthermore, SampEn calculated from sBP beat-domain data failed to reach significance for any combination of *m* and *r* tested (see Figure 2e,f). SampEn calculated from the sBP CIS data was found to be significantly predictive of mortality risk when using *r* ≥ 0.2 for all values of *m*, and the choice of *m* had little effect on mortality prediction above an *r* value of 0.35 (see Figure 2g). SampEn calculated from the dBP CIS data was also significantly predictive of mortality, but only for and *m* values between one and three and *r* values between 0.15 and 0.3 (see Figure 2h). The mean SampEn values and 95% CIs are presented in Figure 3 across the same ranges of *m* and *r* values, stratified by non-deceased and deceased groups. Based on the results presented in Figure 2 and Figure 3, an *r* = 0.3 and *m* = 1 were selected for all subsequent analyses in order to maximize the predictive ability across approaches, while maximizing mean SampEn values and the difference in SampEn values between groups (non-deceased verses deceased). Mean SampEn values (along with 95% CIs) for all *m* values tested are presented as an appendix to this work (see Appendix A, Figure A1).

### 3.3. Associations of Entropy with Mortality Risk

Results from unadjusted Cox proportional hazards regression models examining the associations between baseline sBP and dBP SampEn and all-cause mortality are presented in Figure 4, along with the results for the same models, but using the HRV and RHR values. For the time-domain approach (60 s and 300 s), both sBP and dBP SampEn values were significantly associated with mortality risk, providing HR values between 1.23 and 1.29 per 1 SD (overall 95% CIs = 1.08 to 1.46; *p* ≤ 0.05). SampEn calculated from both sBP and dBP CISs were also associated with mortality risk, with HRs between 1.19 and 1.27 per 1 SD (overall 95% CIs = 1.05 to 1.39; *p* ≤ 0.05). SampEn calculated using the beat-domain approach was not significantly predicative of mortality risk. RHR was positively associated with mortality risk (HR = 1.16 (per 1 SD), 95% CIs = 1.01 to 1.33, *p* ≤ 0.05) and corrected log of SDNN, LF, and HF were all negatively associated with mortality risk (HR = 0.79 (per 1 SD), 95% CIs = 0.69 to 0.91, *p* ≤ 0.05; HR = 0.77 (per 1 SD), 95% CIs = 0.67 to 0.88, *p* ≤ 0.001; HR = 0.81 (per 1 SD), 95% CIs = 0.70 to 0.93, *p* ≤ 0.05, respectively) in these bivariate analyses. Figure 5 presents the results from age-, sex-, and education-adjusted Cox proportional hazards regression models examining the same associations. With the inclusion of age, sex, and education as control variables in the models, the associations between dBP SampEn (CIS approach), and corrected log of SDNN, LF, and HF all lost significances with regards mortality prediction. However, SampEn values calculated from all sBP and dBP time-domain data and sBP CIS data retained significance, though with reduced HR values (see Figure 5). Finally, results from the fully adjusted Cox proportional hazards regression models are presented in Figure 6. In fully adjusted analysis, RHR was not significantly associated with mortality risk. SampEn values calculated from all sBP and dBP time-domain data and sBP CIS data all retained significance in fully adjusted models (see Figure 6). Results (HRs, 95% CIs, and significance levels) for the other variables controlled for in the 60 s time-domain models are presented in Appendix B, Figure A2 as forest plots.

## 4. Discussion

In the present study, we utilized SampEn for the analysis of resting-state BP signal complexity and investigated the associations between these entropy measures and all-cause mortality over a seven-year period in a large sample of community-dwelling older adults in Ireland. We found that higher SampEn in time-domain BP signals was significantly predictive of mortality risk, with an increase of one standard deviation in sBP SampEn and dBP SampEn corresponding to month-on-month HRs of between 1.15 and 1.19, in models comprehensively controlled for other potential confounders. SampEn calculated in time-domain BP data seemed to provide the most robust predictive capability for mortality risk across a range of *m* and *r* values, and it appears that for this purpose, 60 s of resting-state data may be adequate for the measurement of SampEn. SampEn calculated for the sBP CIS data was also predictive of mortality risk in full controlled models, providing a HR of 1.10. Other approaches to SampEn calculated in BP data investigated, as well as more commonly used cardiovascular measures (RHR, log SDNN, log LF and log HF) were not predictive of mortality risk in the analyses.

In previous TILDA studies we demonstrated that higher SampEn measurements from time-domain cardiovascular and neurovascular data were significantly associated with worse pre-disability frailty status, poorer longitudinal cognitive performance, and accelerated brain ageing [2,3,4]. In the present work, we extended this body of evidence, demonstrating that this approach to BP signal entropy measurement is also predictive of all-cause mortality. We suggest that higher signal entropy in continuously measured short-length BP data, via the methodologies reported herein, may be capturing a measure of overall dysregulation of the neurocardiovascular system, as has been previously defined by NCVI [5]. It is plausible that impaired BP control could increase the risk of mortality through reduced/inconsistent blood supply leading to negative alterations in the structure and function of core end-organs such as the heart and the brain. For example, any reduction or interruption in cerebral blood supply can be highly detrimental to the organ’s structure and function, due to the brains limited substrate storage and high metabolic demand [31]. There are several potential physiological causes for higher disorder in BP signals, since several simultaneously active regulatory mechanisms are responsible for short-term cardiovascular control [32,33]. Potential physiological causes for higher BP SampEn include abnormally modified baroreflex sensitivity and/or vagal tone, an increase in sympathetic activity and/or modulation directed to the heart and/or blood vessels, changes in arterial structure (e.g., increased stiffness), modified cardiac reserve, as well as changes of diastolic filling and increased collagen in the left ventricle. In fact, it is most likely that BP SampEn may be influenced by a composite of all the above potential factors. Further work will be required to fully investigate the physiological origins of this novel and potentially clinically useful measure of cardiovascular dysregulation.

In the current study, BtB BP data were used, meaning that absolute sBP and dBP values were only measured at each successive heartbeat. This leads to the question: should entropy calculations be performed on the raw BtB BP data, effectively ignoring the temporal positions of each beat, or should the data be interpolated using both BtB BP and CIS data to retain the time data for each beat, even though this process inevitably decreases the irregularity of the series, and in turn reduces the absolute SampEn values? In this paper we examined the utility of each approach (time-domain BP (interpolated), beat-domain BP, and CIS) and found that the time-domain BP method outperformed the beat-domain BP approach for the prediction of mortality. We propose that this is most likely due to the retention of the time information for the BP data, since short-length heart rate and variability in heart rate (as also captured in the CIS) would also influence the absolute SampEn values with this approach. Additional to this, in our investigation of different SampEn input variables, namely *m* and *r*, we showed that the time-domain approach appeared much less sensitive to the choice of parameters, providing robust, significant results for the predication of mortality across a wide range of values (see Figure 2). This is despite the fact that the *m* and *r* choice affected the mean SampEn values with the time-domain approach more than the other approaches investigated (see Appendix A, Figure A1). Also of note with the time-domain approach used herein, is that it is not simply artificially extending the data by inserting the same number of ‘artificial points’ between beats, a method that has been previously (and correctly) recommended against [13]; but as discussed above, and as in our previous studies [2,3], with this method the temporal positioning of all beats was maintained to within the 5 Hz temporal resolution used, and the number of interpolated points was closely related to the number of heart beats within the time period measured. Also of note with the present study is that although higher RHR on its own seemed associated with a greater risk of mortality, this association was not significant after full adjustment of the models. In the fully adjusted models, several covariates were also associated with mortality prediction (see Appendix B, Figure A2), with age and smoking (current) being positively associated with greater risk, and female sex as well as higher educational attainment being negatively associated, this is in line with previous studies [34,35,36,37]. Being overweight or obese appeared protective against mortality risk (relative to normal weight/underweight), this is again in line with previous studies, which have shown that higher BMI can potentially be protective against all-cause mortality [38].

There are several strengths to the present study, as well as potential avenues for future work. The methodologies used herein are highly transferable for use in a clinical setting, particularly if only one minute of resting state data is required, as this would be very feasible in a busy, time-restricted clinic. Additionally, all measures were non-invasive and non-ionizing, and the equipment pieces required for measuring continuous BP are widely available in a clinical setting. Entropy also provides a single-number measure derived directly from the time series data, allowing for easy interpretation, and tracking over time by clinicians (as long as consistent methodologies are employed with regards SampEn calculation, e.g., for *m* and *r* choice; see Appendix A, Figure A1). Another strength of this study is the large cohort sizes, as well as data being available across a seven-year period. One limitation with the current work is that due to the relatively small number of deaths, only all-cause mortality was investigated. However, it would be informative to explore specific-cause mortality in future work, as this may elucidate some of the underlying physiological mechanisms that BP SampEn is capturing. In addition, despite the population-representative nature of TILDA, replication of this work in independent cohorts will add external validity to our findings.

## 5. Conclusions

This TILDA-based study revealed that higher SampEn in time-domain BP signals was significantly predictive of mortality risk, with an increase of one standard deviation in sBP SampEn and dBP SampEn corresponding to HRs of 1.19 and 1.17, respectively, in models comprehensively controlled for potential confounders. The quantification of SampEn in short length BP signals could provide a novel and clinically useful predictor of mortality risk in older adults.

## Figures and Tables

**Figure 1 entropy-24-00676-f001:**
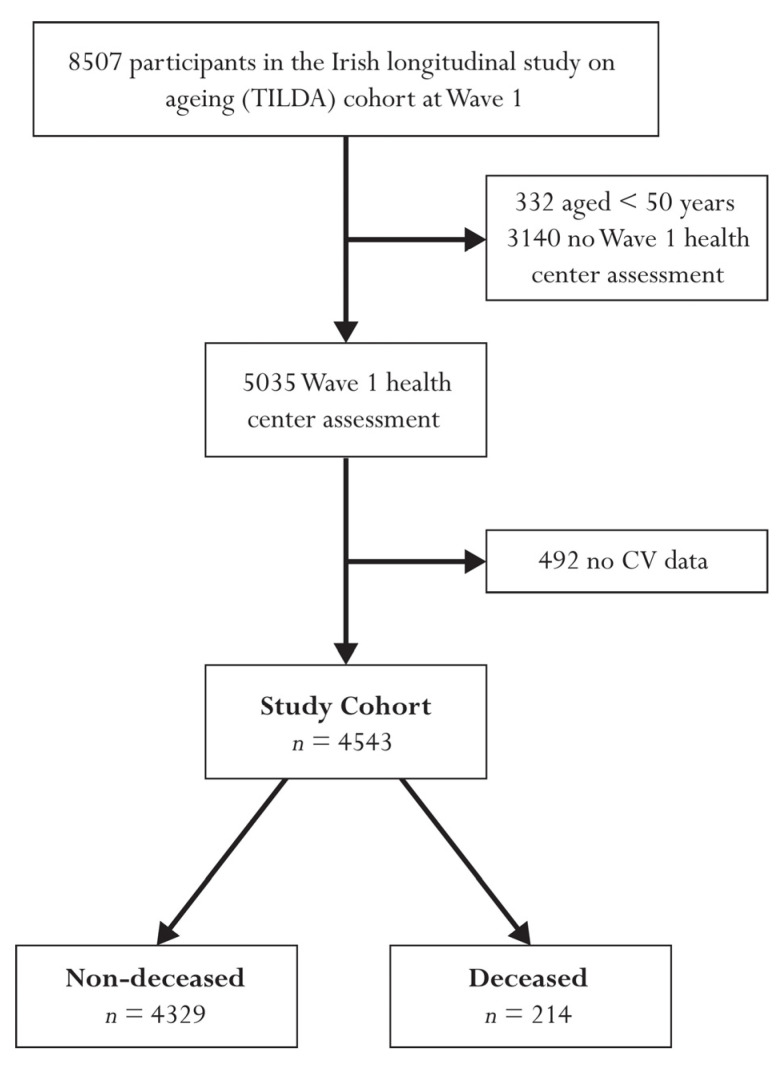
Flow chart describing sample selection and exclusions. Abbreviations: cardiovascular (CV).

**Figure 2 entropy-24-00676-f002:**
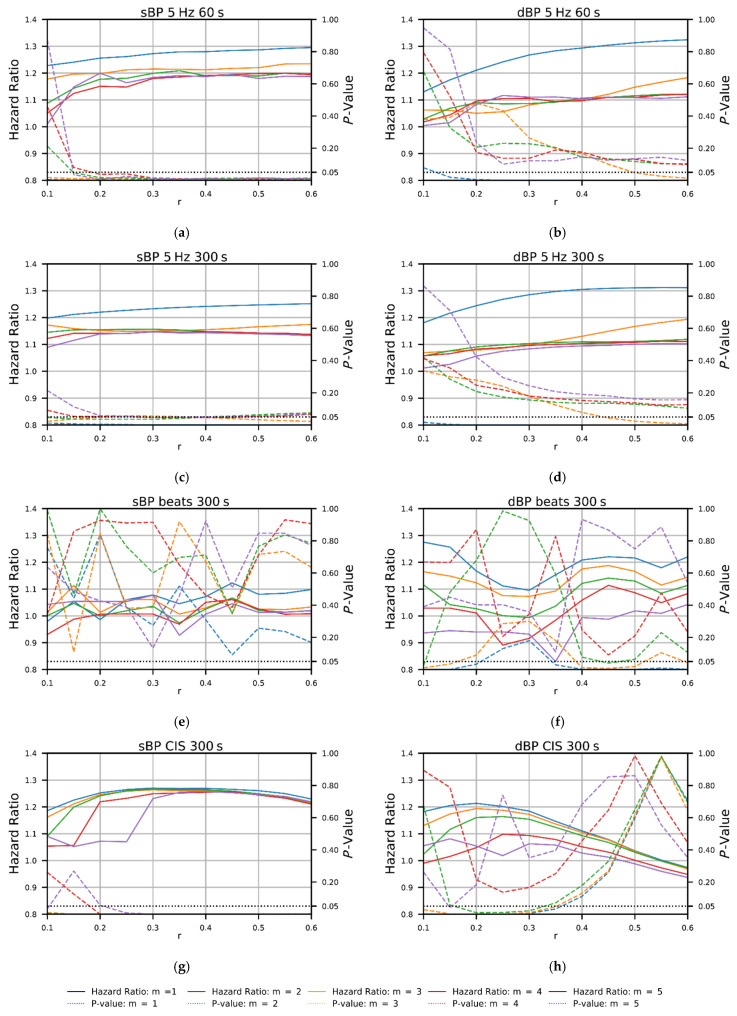
Plots showing the hazard ratios and *p*-values from bivariate Cox proportional hazards regression models investigating associations between sample entropy (SampEn) calculated from time-domain (5 Hz, 60 s and 300 s; **a**–**d**), beat-domain (**e**,**f**), and cardiac interval series (CIS; **g**,**h**) systolic blood pressure (sBP) and diastolic blood pressure (dBP) data and seven-year all-cause mortality, across a range of *m* and *r* values.

**Figure 3 entropy-24-00676-f003:**
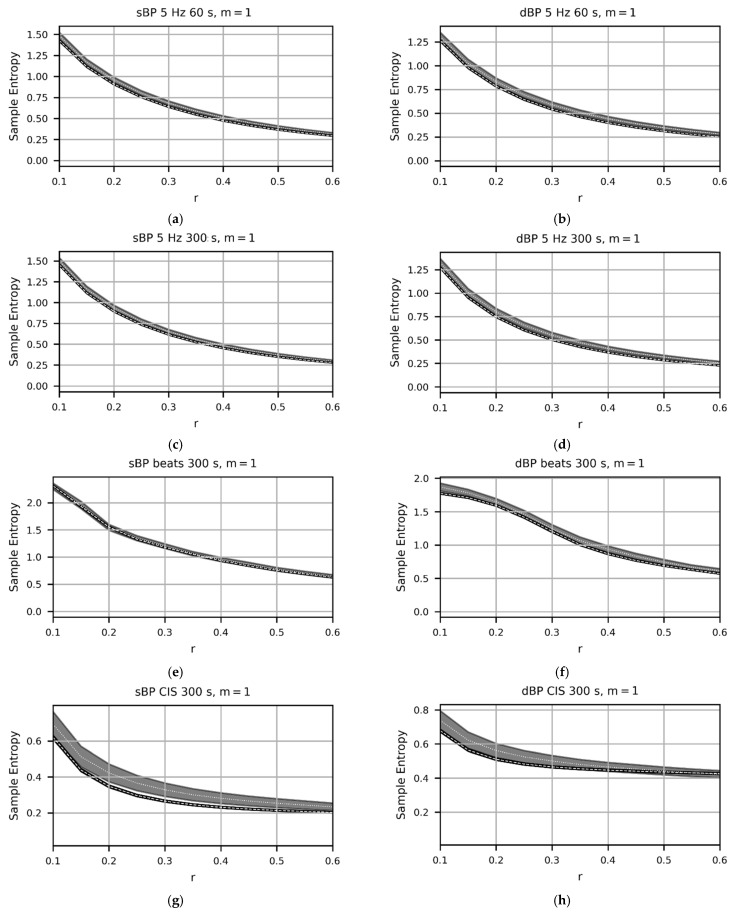
Plots showing sample entropy (SampEn) values calculated using *m* = 1, from time-domain (5 Hz, 60 s and 300 s; (**a**–**d**)), beat-domain (**e**,**f**), and cardiac interval series (CIS; (**g**,**h**)) systolic blood pressure (sBP) and diastolic blood pressure (dBP) data, across a range of *r* values. (Results from the full range of *m* values provided in Appendix A).

**Figure 4 entropy-24-00676-f004:**
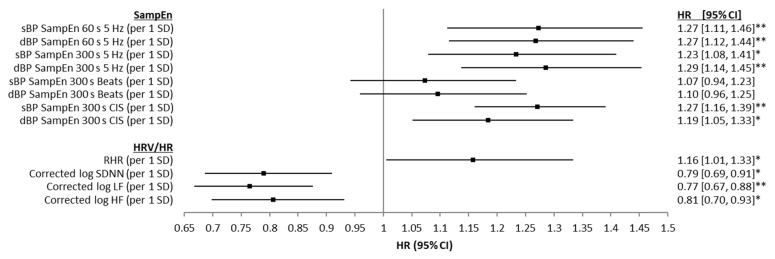
Forest plots showing associations between systolic blood pressure (sBP) and diastolic blood pressure (dBP) sample entropy (SampEn) (for 60 s and 300 s of data, both time-domain and beat-domain), as well as SampEn of cardiac interval series (CIS), resting heart rate (RHR), corrected log of the standard deviation of NN intervals (SDNN), corrected log of low frequency power (LF, 0.04–0.15 Hz), corrected log of high frequency power (HF, 0.15–0.4 Hz) and seven-year all-cause mortality, using unadjusted Cox proportional hazards regression models. Results are presented as z-scored hazard ratios (HR) with 95% CIs for comparability. * *p* ≤ 0.05, ** *p* ≤ 0.001.

**Figure 5 entropy-24-00676-f005:**
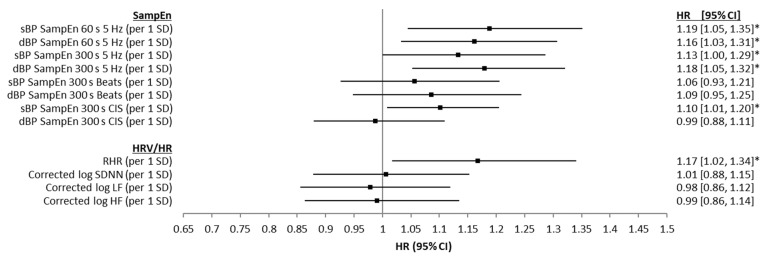
Forest plots showing associations between systolic blood pressure (sBP) and diastolic blood pressure (dBP) sample entropy (SampEn) (for 60 s and 300 s of data, both time-domain and beat-domain), as well as SampEn of cardiac interval series (CIS), resting heart rate (RHR), corrected log of the standard deviation of NN intervals (SDNN), corrected log of low frequency power (LF, 0.04–0.15 Hz), corrected log of high frequency power (HF, 0.15–0.4 Hz) and seven-year all-cause mortality, using age, sex, and education adjusted Cox proportional hazards regression models. Results are presented as z-scored hazard ratios (HR) with 95% CIs for comparability. * *p* ≤ 0.05.

**Figure 6 entropy-24-00676-f006:**
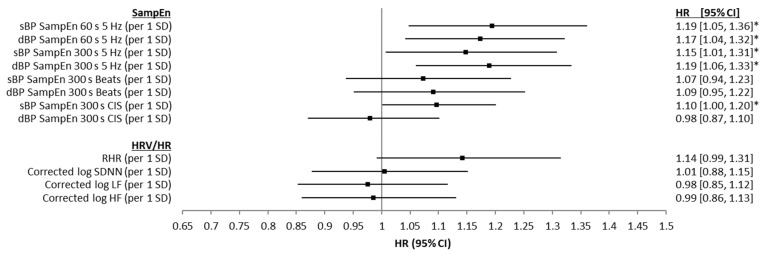
Forest plots showing associations between systolic blood pressure (sBP) and diastolic blood pressure (dBP) sample entropy (SampEn) (for 60 s and 300 s of data, both time-domain and beat-domain), as well as SampEn of cardiac interval series (CIS), resting heart rate (RHR), corrected log of the standard deviation of NN intervals (SDNN), corrected log of low frequency power (LF, 0.04–0.15 Hz), corrected log of high frequency power (HF, 0.15–0.4 Hz) and seven-year all-cause mortality, using fully adjusted Cox proportional hazards regression models. Results are presented as z-scored hazard ratios (HR) with 95% CIs for comparability. All models are fully controlled for the variables outlined in Section 2.6 of the Methods. * *p* ≤ 0.05.

**Table 1 entropy-24-00676-t001:** Baseline demographic characteristics of the study samples. *p* values for differences in demographics between non-deceased and deceased groups, from t-tests and chi-squared tests.

	Full Cohort (*n* = 4543)	Non-Deceased (*n* = 4329)	Deceased (*n* = 214)	*p*
**Age [years]**	61.9 (SD: 8.4, range: [50–91])	61.5 (SD: 8.2, range: [50–90])	70.0 (SD: 9.0, range: [50–91])	**≤0.001**
**Sex [% (*n*)]**	Female: 54.1% (2458)	Female: 54.8% (2371)	Female: 40.7% (87)	**≤0.001**
**Education [% (*n*)]**				**≤0.001**
**Primary/None**	21.5% (977)	20.7% (895)	38.3% (82)	
**Secondary**	41.6% (1890)	41.9% (1814)	29.3% (76)	
**Third/Higher**	36.9% (1676)	37.4% (1620)	26.2% (56)	
**Body Mass Index (BMI) [% (n)]**				0.248
Underweight/Normal BMI	22.9% (1041)	22.7% (980)	28.5% (61)	
Overweight	44.0% (1997)	44.0% (1908)	41.6% (89)	
Obese	23.9% (1086)	24.0% (1041)	21.0% (45)	
Morbidly Obese	9.2% (419)	9.3% (400)	8.9% (19)	
**Antihypertensive Medication Use [% (*n*)]**	33.1% (1503)	32.4% (1401)	47.7% (102)	**≤0.001**
**Self-reported diabetic [%]**	6.5% (295)	6.2% (269)	12.2% (26)	**0.001**
**Number of Cardiovascular Conditions [% (n)]**				**≤0.001**
0	39.3% (1786)	39.6% (1715)	33.2% (71)	
1	34.2% (1555)	34.5% (1494)	28.5% (61)	
2+	26.5% (1202)	25.9% (1120)	38.3% (82)	
Smoker [% (*n*)]				**≤0.001**
Never	45.9% (2084)	46.4% (2010)	34.6% (74)	
Past	39.2% (1784)	39.1% (1693)	42.5% (91)	
Current	14.9% (675)	14.5% (626)	22.9% (49)	
**CAGE Alcohol Scale**				0.461
CAGE < 2	78.1% (3550)	78.3% (3389)	75.2% (161)	
CAGE ≥ 2	12.9% (584)	12.8% (555)	13.6% (29)	
No response	9.0% (409)	8.9% (385)	11.2% (24)	
**SampEn sBP* (60 s 5 Hz)**	0.641 (SD: 0.179, range: [0.022–1.254])	0.639 (SD: 0.178, range: [0.022–1.178])	0.682 (SD: 0.190, range: [0.135–1.254])	**≤0.001**
**SampEn dBP * (60 s 5 Hz)**	0.543 (SD: 0.189, range: [0.024–1.414])	0.540 (SD: 0.187, range: [0.024–1.414])	0.588 (SD: 0.220, range: [0.058–1.227])	**≤0.001**
**SampEn sBP * (300 s 5 Hz)**	0.618 (SD: 0.169, range: [0.074–1.228])	0.616 (SD: 0.168, range: [0.074–1.206])	0.652 (SD: 0.184, range: [0.178–1.228])	**0.002**
**SampEn dBP * (300 s 5 Hz)**	0.505 (SD: 0.176, range: [0.052–1.328])	0.502 (SD: 0.173, range: [0.052–1.328])	0.551 (SD: 0.214, range: [0.138–1.253])	**≤0.001**
**SampEn sBP * (300 s Beats)**	1.179 (SD: 0.291, range: [0.001–2.294])	1.178 (SD: 0.290, range: [0.001–2.294])	1.201 (SD: 0.312, range: [0.239–1.878])	0.272
**SampEn dBP * (300 s Beats)**	1.207 (SD: 0.435, range: [0.001–2.367])	1.205 (SD: 0.435, range: [0.001–2.367])	1.246 (SD: 0.434, range: [0.368–2.266])	0.179
**SampEn sBP * (300 s CIS)**	0.268 (SD: 0.174, range: [0.001–1.594])	0.265 (SD: 0.166, range: [0.001–1.594])	0.329 (SD: 0.278, range: [0.076–1.568])	**≤0.001**
**SampEn dBP * (300 s CIS)**	0.468 (SD: 0.177, range: [0.001–1.534])	0.466 (SD: 0.173, range: [0.001–1.534])	0.501 (SD: 0.239, range: [0.128–1.526])	**0.005**
**RHR [bpm]**	64.0 (SD: 10.1, range: [37.6–117.2])	63.9 (SD: 9.9, range: [37.6–111.7])	65.4 (SD: 12.3, range: [40.8–117.2])	**0.047**
**HRV SDNN ** [ms]**	37.8 (SD: 17.8, range: [1.8–164.9])	37.9 (SD: 17.8, range: [1.8–164.9])	34.2 (SD: 16.5, range: [3.1–108.2])	**0.006**
**HRV LF ** [ms^2^]**	427.9 (SD: 579.3, range: [0.2–9152.3])	431.8 (SD: 582.8, range: [0.5–9152.3])	340.9 (SD: 486.8, range: [0.2–3734.5])	**0.041**
**HRV HF ** [ms^2^]**	218.8 (SD: 310.5, range: [0.5–4109.5])	219.8 (SD: 311.7, range: [0.5–4109.5])	196.1 (SD: 282.4, range: [1.2–1994.4])	0.318

* *r* = 0.3; *m* = 1. ** Non-corrected. Abbreviations: systolic blood pressure (sBP); diastolic blood pressure (dBP); sample entropy (SampEn); cardiac interval series (CIS); resting heart rate (RHR); beats-per-minute (bpm); heart rate variability (HRV); standard deviation of NN intervals (SDNN); low frequency power (LF); high frequency power (HF). Bold *p* values indicate significance was reached (*p* ≤ 0.05).

## Data Availability

The datasets generated during and/or analyzed during the current study are not publicly available due to data protection regulations but are accessible at TILDA on reasonable request. The procedures to gain access to TILDA data are specified at https://tilda.tcd.ie/data/accessing-data/, (accessed on 12 October 2021).

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
