# Peer review of "Cardiovascular Signal Entropy Predicts All-Cause Mortality: Evidence from The Irish Longitudinal Study on Ageing (TILDA)"

_entropy, 2022, doi:10.3390/e24050676_

Round 1

Reviewer 1 Report

The study uses sample entropy of blood pressure to study the predictability of the death in a large longitudinal study over seven-year period. The number of deaths was 214 of the total sample of 4543 participants. Higher sample entropy of the five-minute resting state systolic and diastolic blood pressure, indicating low predictability over samples, was predictive of higher mortality risk. This effect was significant after controlling several confounding factors. The sample entropy has potential for clinical use because the instrumentation is widely available, and the computation of sample entropy is simple. However, to study the specific causes of mortality another study will be needed in the future.

I do not have any suggestions for the improvement of the manuscript.

Author Response

We thank the reviewer for taking the time to review our paper. We have now implemented major revisions at the request of Reviewer 2. The three main changes we have made are:

  1. We have changed the language used to describe the different approaches to data processing (‘time-domain’ and ‘beat-domain’) to avoid confusion.
  2. We now include associations between SampEn measured from the cardiac interval series (CIS) data and all-cause mortality
  3. We now report results across a range of m and r values, and this has informed the use of different SampEn input parameter values for the main result, optimized to maximise prediction of mortality.

However, this has not changed the main message or results of the paper.

Reviewer 2 Report

The paper from Knight et. al. addresses an interesting topic, which is studied in a large population. However, in my opinion, many points of the text should be clarified, analyses should be included/excluded and the methodology should be revised.

General points

  • Since SampEn is little explored for BP signals, it would be useful if the authors could evaluate different “m” and “r” parameters and see how the stratification of mortality changes with them.
  • I do not see a compelling rationale for interpolating the series in this study. As the authors already noticed, when you interpolate for 5 Hz, the timing between each sample is way shorter compared to the original BtB series, introducing spurious regularity in the series. The difference noticed by the authors in the two examples in the Appendix is actually caused by the different mean heart rates. I would expect a high association between mean heart rate and the SampEn calculated from interpolated series. Moreover, the hazard ratios for BtB and interpolated series are not much different, as presented in Figure 4. Thus, I suggest the authors remove the analysis with interpolated series and keep only BtB series, which is the common practice.
  • SampEn is widely applied to cardiac interval series and it would be very informative to see its predictive value compared to SampEn of BP, no matter which one gives the highest predictive value. So, please include SampEn of the cardiac interval series.
  • Many time- and frequency-domain HRV indices are recognizably influenced by the mean heart rate (see, e.g. DOI:10.1111/anec.12148). Therefore, I suggest the authors include corrected SDNN and corrected spectral powers in their prediction analyzes.

Specific points

Abstract

Please, include the age range of the population evaluated.

Introduction

  • Line 52: although measures of NCVI based on blood pressure dynamics are, indeed, lacking, there is plenty of literature on it using HRV series. Please, revise this sentence.
  • Lines 66-68: I did not understand what you mean by “different ways to calculate SampEn”. Are you referring to the choice of parameters? Please, clarify.
  • Line 68: “… to know how SampEn of BP signals compares with…”
  • Line 74: again here, what do you mean by “different approaches to the calculation of SampEn”?
  • Although scarce, there are some papers addressing the analysis of entropy in BP. Please, provide a brief summary of what has been studied in this regard.

Materials and Methods

  • Why did the authors choose 200 Hz for BP recordings? It is too low for beat-to-beat variability analysis of humans and should be considered a limitation of the study. Look at the discrepancy compared to the ECG recordings (4 kHz).
  • Line 97: did you allow a few minutes of stabilization before the recordings start?
  • Lines 100-101: I did not understand this sentence. BtB series are not evenly spaced in time.
  • Line 105-109: I did not understand this sentence as well. How can you keep the original beat positions after interpolating the series? You will naturally create the “artificial” points in the interpolated series, even though they are based on the original ones. Please, elaborate more on it.
  • Lines 120-126: one cannot understand what the template vectors are only from this sentence. You must first define the time series from which these vectors are obtained and more clearly state that those vectors are m-length, i.e. range from j to j+m-1.
  • Line 180: “different entropy approaches”. Which approaches? It is not clear.

Results

  • Line 199: what do you mean by “seamed higher”? Didn’t you statistically compare “deceased” vs “non-deceased” groups? If not, how can you say the deceased cohort is older, less educated, with a higher portion of hypertensive use, etc? Please, include the corresponding appropriate statistical tests for each variable in Table 1.
  • Line 247: lower? Is it not the opposite?
  • Figures 2, 3 and 4: I am not sure what the HR of covariates are. I understand that you create models for SampEn corrected for those covariates. But did you also create models for each covariate even when the model is correct for them?

Discussion

  • Lines 319-321: again here, this is not clear how you can keep the beat position after interpolating it to 5 Hz.
  • Lines 321-323: where are the results for unadjusted models with RHR?
  • Lines 266-268: hazard ratios for SampEn are not the highest. Smoking history seems to be the highest predictor.
  • Lines 314-315: “how a BtB approach may sometimes remove important information from the BP data”. I don’t think the interpolated series has “more information”, since the interpolation is derived from the real values, so no additional information is actually obtained.

Author Response

General points

  • Since SampEn is little explored for BP signals, it would be useful if the authors could evaluate different “m” and “r” parameters and see how the stratification of mortality changes with them.

Response: We thank the reviewer for their input here and have now included results for all combinations of m = [1, 2, 3, 4, 5] and r = [0.1, 0.15, 0.2, 0.25, 0.3, 0.35, 0.4, 0.45, 0.5, 0.55, 0.6] using bivariate analysis in the paper. This has provided some interesting new results and has indeed increased the predictive power of SampEn with the time-domain data. This new addition will hopefully also prove useful with regards informing methodologies for future studies.

  • I do not see a compelling rationale for interpolating the series in this study. As the authors already noticed, when you interpolate for 5 Hz, the timing between each sample is way shorter compared to the original BtB series, introducing spurious regularity in the series. The difference noticed by the authors in the two examples in the Appendix is actually caused by the different mean heart rates. I would expect a high association between mean heart rate and the SampEn calculated from interpolated series. Moreover, the hazard ratios for BtB and interpolated series are not much different, as presented in Figure 4. Thus, I suggest the authors remove the analysis with interpolated series and keep only BtB series, which is the common practice.

Response: Reading through the reviewer’s other comments below, we realise that perhaps we may not have been clear enough about our rationale for the ‘interpolated’ time-domain approach and, respectfully, we would like to take the opportunity to further elaborate here.

When extracting BtB data from the raw BP waveform measured using the Finometer device, one extracts two vectors, the BtB BP values, sBP (the peaks of the raw BP waveform) and dBP (the troughs of the raw waveform), and the corresponding BtB cardiac interval series (CIS) values for each. SampEn values are calculated from a vector of data, on the assumption that all points on said vector are evenly spaced. Therefore, we can either calculate SampEn for the BtB ‘beat-domain’ BP values, with each ‘beat’ representing a consecutive data point on that vector (therefore ignoring where the data points are located in the time domain) OR on the BtB CIS values (ignoring the absolute BP values at each timepoint). An alternative approach (as we propose in this work) is to interpolate the data using both the BtB BP and CIS data, therefore arriving at a new vector wherein both absolute BP values AND their position in the time domain have been considered, for which we can then calculate SampEn.

We now understand that we may have explained this in a confusing way in the initial draft of the paper and as such have adopted different nomenclature throughout the manuscript to describe these two approaches to data processing prior to the SampEn calculation, namely ‘beat domain’ BP and 5Hz ‘time domain’ BP. We hope that these amendments, coupled with the explanation above, clarify our two approaches to BP data processing. We have also included results from SampEn calculated from the CIS data.

We respectfully disagree with the reviewer that we should remove the ‘time domain’ result and just report the ‘beat domain’ data as our main results. The basic idea behind using this approach (time domain) for the calculation of SampEn was that it takes into account a composite of relevant cardiovascular parameters, namely BP variability (as could be quantified using ‘beat domain’ SampEn), CIS variability (as could be quantified using CIS SampEn), and RHR.

We take the reviewer’s comment with regards the examples in the Appendix, and we indeed mention the influence of RHR on time-domain SampEn in the Discussion. We have now replaced Figure A1 with a more comprehensive (and hopefully less confusing) figure which reports graphically all mean SampEn values with 95% CIs for all approaches investigated in this work, stratified by deceased and non-deceased groups across the range of r and m values investigated.

We trust that our responses to the reviewer’s individual concerns below and accompanying amendments to the manuscript clarify our rationale in taking this approach. We thank the reviewer for highlighting this and allowing us the opportunity to clarify our rationale and methodology.

  • SampEn is widely applied to cardiac interval series and it would be very informative to see its predictive value compared to SampEn of BP, no matter which one gives the highest predictive value. So, please include SampEn of the cardiac interval series.

Response: We have now included the results for SampEn of the CIS versus mortality prediction in the paper, with some interesting results. We thank the reviewer for their input on this point.

  • Many time- and frequency-domain HRV indices are recognizably influenced by the mean heart rate (see, e.g. DOI:10.1111/anec.12148). Therefore, I suggest the authors include corrected SDNN and corrected spectral powers in their prediction analyzes.

Response: We have now included corrected SDNN and spectral power measures in the manuscript. We have also included a reference to the suggested paper. We thank the reviewer for this suggestion.

Specific points

Abstract

Please, include the age range of the population evaluated.

Response: Age and sex demographics are now included in the abstract.

Introduction

  • Line 52: although measures of NCVI based on blood pressure dynamics are, indeed, lacking, there is plenty of literature on it using HRV series. Please, revise this sentence.

Response: Sentence amended to: “However, measures of NCVI based on BP that capture this risk are lacking.”

  • Lines 66-68: I did not understand what you mean by “different ways to calculate SampEn”. Are you referring to the choice of parameters? Please, clarify.

Response: Sentence amended to: “However, for health research applications, different ways of processing continuously measured beat-to-beat BP signals prior to SampEn calculation may yield different results. Previous works have investigated SampEn calculations from ‘beat-domain’ BP [2-4, 13], ‘time-domain’ BP (incorporating both BtB BP and cardiac interval series (CIS) data via interpolation) [2-4], and CIS approaches [13-15]; in present work we aimed to investigate the utility of all three approaches with regards mortality prediction.”

  • Line 68: “… to know how SampEn of BP signals compares with…”

Response: Amended, thank you.

  • Line 74: again here, what do you mean by “different approaches to the calculation of SampEn”?

Response: Sentence amended to: “In doing so, we compared different processing approaches for BtB BP data prior to the calculation of SampEn (time-domain at 5Hz, beat-domain, and CIS), we investigated these approaches across a range of m and r values, and also compared the ability of these different entropy measures to predict mortality with HRV and RHR measures.”

  • Although scarce, there are some papers addressing the analysis of entropy in BP. Please, provide a brief summary of what has been studied in this regard.

Response: Sentence added: “Previous works have investigated SampEn calculations from ‘beat-domain’ BP [2-4, 13], ‘time-domain’ BP (incorporating both BtB BP and cardiac interval series (CIS) data via interpolation) [2-4], and CIS approaches [13-15]; in present work we aimed to investigate the utility of all three approaches with regards mortality prediction.”

Materials and Methods

  • Why did the authors choose 200 Hz for BP recordings? It is too low for beat-to-beat variability analysis of humans and should be considered a limitation of the study. Look at the discrepancy compared to the ECG recordings (4 kHz).

Response: We respectfully disagree with the reviewer on this point, since these devices record different physiological measures. The Finometer device is measuring the raw BP waveform (finger volume pulse waveform), and for this we regard 200Hz as more than adequate since we aim to extract the waveform peaks (sBP) and troughs (dBP), which occur at each heartbeat. Considering the highest RHR from this study (117 bpm, i.e. 1.95 Hz), a sampling frequency of 200Hz is more than adequate to capture this data. ECG detects the electrical signals produced by the heart each time it beats and as such requires much higher sampling rates for accurate quantification.

  • Line 97: did you allow a few minutes of stabilization before the recordings start?

Response: These data were acquired continuously as part of a comprehensive cardiovascular examination, which included 5 min free breathing during supine rest (the data presented in the present work), 5 mins paced breathing during supine rest, and an active stand / recovery from stand. Most individuals would have laid supine for 1-2 mins prior to the 5-minute section of data used in the present study being recorded. We clarify in the paper:

 “Participants laid supine, and after a stabilization period five minutes of data were collected.”

  • Lines 100-101: I did not understand this sentence. BtB series are not evenly spaced in time.

Response: We hope that our response to the second general point above, coupled with the now amended text in the manuscript, clarifies this point.

  • Line 105-109: I did not understand this sentence as well. How can you keep the original beat positions after interpolating the series? You will naturally create the “artificial” points in the interpolated series, even though they are based on the original ones. Please, elaborate more on it.

Response: Again, we hope that our response to the second general point above, coupled with the now amended text in the manuscript, clarifies this point.

  • Lines 120-126: one cannot understand what the template vectors are only from this sentence. You must first define the time series from which these vectors are obtained and more clearly state that those vectors are m-length, i.e. range from j to j+m-1.

Response: We have now clarified this point in the manuscript as:

“For a time series of length N, Bmi(r) is defined as the number of template vectors of length m, xm(j), similar to xm(i) (within r) divided by Nm − 1, where j = 1...Nm, with ji (to avoid self-matches). The average Bmi(r) for all i is given as

EQ1

(1)

Similarly, we define Ami(r) as the number of template vectors of length m + 1,  xm+1(j), similar to  xm+1(i) (within r) divided by Nm − 1, where j = 1...Nm, with ji. The average Ami(r) for all i is given as

EQ2

(2)

SampEn was then calculated as

EQ3

(3)

  • Line 180: “different entropy approaches”. Which approaches? It is not clear.

Response: This is now clarified in the manuscript.

Results

  • Line 199: what do you mean by “seamed higher”? Didn’t you statistically compare “deceased” vs “non-deceased” groups? If not, how can you say the deceased cohort is older, less educated, with a higher portion of hypertensive use, etc? Please, include the corresponding appropriate statistical tests for each variable in Table 1.

Response: We thank the reviewer for their comment and have now clarified this point with the inclusion of statistical test results in Table 1, and the corresponding updated text:

“The deceased cohort were less educated, had a higher proportion of antihypertensive medication use, had almost twice the prevalence of diabetes, had a greater number of cardiovascular conditions, and had a higher proportion of past or current smokers. Average SampEn measures were significantly higher in the deceased cohort, compared with the non-deceased, for both time-domain and CIS approaches, however, not for the beat-domain approach to data processing.”

  • Line 247: lower? Is it not the opposite?

Response: Thank you for your question but it is in fact not the opposite. Previous meta-analyses have shown that higher BMI can potentially be protective against all-cause mortality (https://www.ncbi.nlm.nih.gov/pmc/articles/PMC4855514/). This point has been added to the Discussion section:

“Being overweight or obese appeared protective against mortality risk (relative to normal weight / underweight), this is again in line with previous studies which have shown that higher BMI can potentially be protective against all-cause mortality [38].”

  • Figures 2, 3 and 4: I am not sure what the HR of covariates are. I understand that you create models for SampEn corrected for those covariates. But did you also create models for each covariate even when the model is correct for them?

Response: HRs for covariates are derived all from the same model. To avoid confusion, and not to detract from the main message of the paper, these results have now been moved to Appendix B. We have also removed reference to these HRs from the Results section and moved this to a Discussion point, with the following text:

“In the fully adjusted models, several covariates were also associated with mortality prediction (see Appendix B, Figure B1), with age and smoking (current) being positively associated with greater risk, and female sex as well as higher educational attainment being negatively associated, this is in line with previous studies [34-37]. Being overweight or obese appeared protective against mortality risk (relative to normal weight / underweight), this is again in line with previous studies which have shown that higher BMI can potentially be protective against all-cause mortality [38].”

Discussion

  • Lines 319-321: again here, this is not clear how you can keep the beat position after interpolating it to 5 Hz.

Response: Once again, we hope that our response to the second general point above, coupled with the now amended text in the manuscript, clarifies this point.

  • Lines 321-323: where are the results for unadjusted models with RHR?

Response: Results from unadjusted; age-, sex-, and education-adjusted; as well as fully adjusted models are now included in the main results of the study.

  • Lines 266-268: hazard ratios for SampEn are not the highest. Smoking history seems to be the highest predictor.

Response: HR results from the fully controlled model covariates have now been moved to an Appendix and reference to these results has been removed from the Results section (however, we have added this as a Discussion point) to avoid confusion.

  • Lines 314-315: “how a BtB approach may sometimes remove important information from the BP data”. I don’t think the interpolated series has “more information”, since the interpolation is derived from the real values, so no additional information is actually obtained.

Response: Once more, we hope that our response to the second general point above, coupled with the now amended text in the manuscript, clarifies this point.

Round 2

Reviewer 2 Report

I would like to thank the authors for their detailed responses to my comments. In general, I believe the quality of the paper improved considerably.

I still disagree with the authors' point of view regarding the use of interpolated BP and CI series. SampEn, differently from spectral analysis, for example, does not depend on the time the beats occur. Actually, it is intended to extract the irregularity on a beat-by-beat basis, which accounts for the physiological mechanisms varying within one cardiac cycle.

On the other hand, I respect the authors' opposite opinion and I believe the paper is sound enough to be published. The relevance of the interpolated approach should be judged by the scientific community, which is one of the purposes of scientific communications.

As a final suggestion, the authors could create a scatter plot of RHR vs SampEn in the two approaches ("beat to beat" and "time domain"). No matter if the authors chose to include it or not, I recommend the paper publication.